# An Enhanced Indoor Three-Dimensional Localization System with Sensor Fusion Based on Ultra-Wideband Ranging and Dual Barometer Altimetry [note 1]

**DOI:** 10.3390/s24113341

**Published:** 2024-05-23

**Authors:** Le Bao, Kai Li, Joosun Lee, Wenbin Dong, Wenqi Li, Kyoosik Shin, Wansoo Kim

**Affiliations:** 1Department of Mechatronics Engineering, Hanyang University, Ansan 15588, Republic of Korea; baole@hanyang.ac.kr (L.B.); likai@hanyang.ac.kr (K.L.); km01049@hanyang.ac.kr (J.L.); wuhui0514@hanyang.ac.kr (W.D.); liwenqi@hanyang.ac.kr (W.L.); 2Robotics Department, Hanyang University ERICA, Ansan 15588, Republic of Korea; norwalk87@hanyang.ac.kr

**Keywords:** indoor localization, Kalman filter, ultra-wideband sensor, barometric pressure sensor

## Abstract

Accurate three-dimensional (3D) localization within indoor environments is crucial for enhancing item-based application services, yet current systems often struggle with localization accuracy and height estimation. This study introduces an advanced 3D localization system that integrates updated ultra-wideband (UWB) sensors and dual barometric pressure (BMP) sensors. Utilizing three fixed UWB anchors, the system employs geometric modeling and Kalman filtering for precise tag 3D spatial localization. Building on our previous research on indoor height measurement with dual BMP sensors, the proposed system demonstrates significant improvements in data processing speed and stability. Our enhancements include a new geometric localization model and an optimized Kalman filtering algorithm, which are validated by a high-precision motion capture system. The results show that the localization error is significantly reduced, with height accuracy of approximately ±0.05 m, and the Root Mean Square Error (RMSE) of the 3D localization system reaches 0.0740 m. The system offers expanded locatable space and faster data output rates, delivering reliable performance that supports advanced applications requiring detailed 3D indoor localization.

## 1. Introduction

Indoor localization technology is essential for enabling precise personnel tracking and efficient robot navigation in complex environments [1,2,3]. Its applications significantly improve efficiency and convenience [4]. Indoor localization can be divided into two categories according to the size of the localization area. The first category addresses larger areas, such as halls or multi-story buildings [5,6]. This scenario often requires multiple anchor points as references [7], or multi-sensor fusion [8]. The localization requirements are meter-level plane localization, and floor identification. The second category focuses on high-precision localization within a slightly smaller indoor area. Some existing technologies have already achieved indoor two-dimensional (2D) localization, for example, visual features [9], Light Detection and Ranging (LiDAR) [10,11], and wireless localization [12]. However, with the advancement of indoor three-dimensional (3D) localization sensor technology, emerging services like indoor drone control [2,13], virtual reality (VR) [14,15], or augmented reality (AR) [16] experiences demand higher accuracy in location. They also require more stringent height estimation and more frequent updates of information [17,18]. Additionally, while exploring the potential of these technologies, it is important to consider the cost factor [19] to ensure that technological solutions have the potential for widespread application. Based on our previous research [20], this study proposes an enhanced indoor 3D localization system, which fuses ultra-wideband (UWB) sensor ranging and barometric pressure (BMP) sensor height measurement.

Commonly used wireless sensors for ranging and localization include Wi-Fi, Bluetooth, and Radio Frequency Identification (RFID) [21]. They estimate the target’s location by measuring signal strength or time differences [1], with meter-level accuracy. Although they satisfy commercial and industrial needs to a certain extent, they still face challenges in positioning accuracy and stability in complex indoor environments. Additionally, visual positioning systems and LiDAR offer higher accuracy but their implementation costs and demands on computational resources limit their potential for widespread application [9,10]. In the realm of indoor localization, technology stands out for its high precision in distance measurement by transmitting ultra-short pulse signals [22]. This method achieves centimeter-level ranging accuracy, surpassing traditional Wi-Fi and Bluetooth technologies in both accuracy and interference resistance [3,23,24]. The target’s 3D location can be calculated based on the distance measurement information between the target and each UWB base station. However, despite the superior distance measurement capabilities of UWB, existing research indicates persistent errors in height estimation [13], underscoring the need for continued research to optimize localization algorithms and enhance system performance.

In contrast, barometric pressure sensors (BMPs) enhance height measurement accuracy [25]. The earliest mercury barometer was invented in the 17th century by physicist Evangelista Torricelli [26]. Thanks to ongoing technological advancements, modern electronic barometers [27], which convert barometric pressure into electrical signals, are widely used. These BMP sensors can precisely measure barometric pressure and output digital signals, thereby estimating the corresponding height. By measuring atmospheric pressure changes, BMP sensors provide critical data for adjusting the vertical positioning in indoor localization systems, which is often flawed in UWB-only setups. However, the barometric pressure value at a given location can vary over time [6]. Outdoors, the shifting of high- and low-pressure weather systems can alter barometric pressure [28]. Especially strong convective weather accelerates changes in barometric pressure, leading to increased height drift errors [29]. Additionally, changes in air temperature cause air expansion or contraction, resulting in decreased or increased barometric pressure. In indoor environments, factors such as the building’s sealing, ventilation conditions, and the temperature difference between indoors and outdoors predominantly influence barometric pressure [30]. The barometric pressure in well-sealed indoor environments is relatively stable. In contrast, the opening and closing of doors and windows, along with the ventilation system, can cause fluctuations in indoor barometric pressure, which limits the single BMP sensor’s performance for height estimation in indoor environments.

Current indoor localization systems often fail to deliver accurate height measurements, a gap our system addresses by integrating BMP sensors known for their reliable barometric pressure measurements. This integration ensures indoor localization system enhances not only vertical but also horizontal localization precision. Based on our previous research [20], we have upgraded the hardware and software. By observing the trend of barometric pressure changes at different indoor locations over 30 min using BMP sensors, we confirmed once again that the drift of barometric pressure in the same indoor space tends to be consistent. In this study, we introduced an enhanced indoor 3D localization system. The barometric pressure values at the tag are transmitted to the main controller via Wi-Fi signals. The relative height between the tag and anchor is then calculated using derived formulas. The system’s main controller employs a dual-core processor with Wi-Fi capability, enhancing the rate and stability of data transfer and processing. The height measurements obtained from the dual BMP sensors are used to assist the UWB sensors in spatial ranging. The distance measurements from the UWB sensors are refined and corrected using a fitting equation. The localization system includes three anchors at the same height, where UWB sensors on each anchor measure the distances to the tag. The tag’s planar position is estimated through geometric modeling and the centroid method of triangles. Finally, Kalman filtering is applied to obtain the most accurate estimate of the tag’s location, resulting in a smoother and more precise localization trajectory. Compared to previous research, the new hardware support and software architecture optimization have enabled the system to obtain sensor information more quickly and stably. The dispersed anchor setup allows for spatial localization over a larger area, and the redesigned geometric localization model and Kalman filtering algorithm have further improved localization accuracy.

Relative to similar localization strategies [13,31], which typically exhibit height estimation errors exceeding 0.2 m, our dual BMP sensors-based method significantly improves accuracy in height estimation, thus enhancing the system’s 3D localization precision. This work’s main contributions are as follows:Proposed and validated a method for estimating tag height based on dual BMP sensors, effectively compensating for most indoor barometric pressure deviations, and providing a more accurate height estimation than achievable with a single BMP sensor. For the challenge of indoor height estimation, our accuracy is approximately ±0.05 m, with an RMSE of 0.0282 m.Developed a hardware framework that enhances the system to be more efficient and stable, with a localization output rate reaching 37 Hz, a nine-fold increase compared to earlier designs.Our portable localization system covers a larger measurable range. The proposed geometric localization model and the Kalman filtering technique are empirically validated, showing a 2D localization RMSE of 0.0585 m, and a 3D localization RMSE of 0.0740 m. Compared to indoor localization systems with a similar number of anchors, ours offers an extended measurable range and superior accuracy.

## 2. Hardware System Design

The hardware system framework is depicted in Figure 1. The target (Tag) contains one UWB sensor and one BMP sensor. The indoor localization system features three anchors (Anchor (A), Anchor (B), Anchor (C)), each equipped with three UWB sensors and one BMP sensor to pinpoint the target’s (Tag) location. The main controller, ESP32 (manufactured by Espressif Systems, Shanghai, China) is responsible for receiving data from the sensors, calculating the tag’s location, and sending critical data to an external computer for data collection. The distances between the tag and each anchor are estimated by UWB technology. The ranging results are collected by the sub-controller of Anchor (A) and transmitted to the main controller via I2C communication. The BMP sensor on the tag measures the barometric pressure in real time, which is collected and processed by an ESP8266 controller (manufactured by Espressif Systems, China), and then sent to the main controller by Wi-Fi wireless communication. Furthermore, thanks to the dual-core processor of the main controller (ESP32), the Wi-Fi signal-reading process and the location calculation process do not interfere with each other, greatly enhancing the stability of the localization system.

Figure 2 shows the tag’s hardware design. The tag primarily consists of a controller (ESP8266), a BMP sensor, and a UWB sensor, all powered by a shared 5 V battery. The controller (ESP8266) and the BMP sensor are soldered onto a printed circuit board (PCB), with their I2C ports connected by the internal circuit. Finally, these components are assembled in a 3D-printed plastic frame. Additionally, a marker for the motion capture system (MCS) is also mounted on the top of the frame, which can be used for experimental validation. The anchors of the localization system are mounted on three tripods, which are powered by either a computer or batteries as shown in Figure 3. The UWB sensors in each anchor need to be adjusted to the same height using the tripods. Also, the proposed localization system is easy to set up in new indoor environments due to the portability of the tripods. And the specific parameters of the hardware devices in the designed system are detailed in Table 1.

## 3. Spatial Distance Measurement

### 3.1. Height Measurement with Dual BMP Sensors

The barometric pressure varies with changes in altitude and can drift over time. BMP sensors estimate height by measuring barometric pressure values. In a previous study [20], we proposed a method for estimating indoor target height using dual BMP sensors. In this new study, we have revisited the observation of barometric pressure variations at fixed indoor locations, and conducted re-validation in the updated localization system.

#### 3.1.1. Observation of Indoor Barometric Pressure Measurements

In an office indoor environment, we collected barometric pressure data using two BMP sensors. The BMP sensor of Anchor (A) remained stationary, and initially, the tag’s BMP sensor was positioned close to Anchor (A) at the same height. A few seconds after initiating the timer, the tag moved to another location and remained stationary, with a horizontal distance of 2 m and a height drop to −0.5 m. This observation lasted for 30 min (i.e., 1800 s). Due to the significant noise in the raw measurement data from the BMP sensor, a weighted average filter was used to minimize the noise. The results of this observation are displayed in Figure 4.

The observation results show that there is a continuous drift in the indoor barometric pressure values over thirty minutes. After filtering, the data noise from the BMP sensors was reduced. However, even if the two sensors are of the same model, the barometric pressure values they measure at the same height location will still have minor differences $P_{diff}$. To unify this deviation, the final output barometric pressure value of the tag $P_{Tag(t)}^{\prime }$ was adjusted by adding $P_{diff}$ as the following equation:(1)$$\begin{array}{cc} \; & P_{diff}=P_{Anchor(0)}-P_{Tag(0)}\;, \end{array}$$(2)$$\begin{array}{cc} \; & P_{Tag(t)}^{\prime }=P_{Tag(t)}+P_{diff}\;, \end{array}$$
where $P_{Anchor(0)}$ is the initial barometric pressure value of Anchor (A), $P_{Tag(0)}$ is the initial barometric pressure value of the tag, and $P_{Tag(t)}$ is the barometric pressure value of the tag measured at the current moment. After the tag’s location was lowered, its barometric pressure value increased. During the observed 30 min, the trends of barometric drift at the two stationary positions were almost identical.

#### 3.1.2. Relative Height Estimation Based on Dual BMP Sensors

After obtaining the final barometric pressure data from the BMP sensors ($P_{Tag(t)}^{\prime }$ and $P_{Anchor(t)}$), the height values of the tag and the anchor ($H_{Tag(t)}$ and $H_{Anchor(t)}$) can be determined by the barometric height calculation [32] as follows: (3)$$\begin{array}{cc} H_{Anchor(t)} & =\left[1-{\left(\frac{P_{Anchor(t)}}{P_{0}}\right)}^{\frac{1}{5.257}}\right]\cdot \left(\frac{T+273.15}{0.0065}\right)\;, \end{array}$$(4)$$\begin{array}{cc} H_{Tag(t)} & =\left[1-{\left(\frac{P_{Tag(t)}^{\prime }}{P_{0}}\right)}^{\frac{1}{5.257}}\right]\cdot \left(\frac{T+273.15}{0.0065}\right)\;, \end{array}$$
where the reference barometric pressure $P_{0}$ is equal to Anchor (A)’s initial value $P_{Anchor(0)}$. The *T* is the temperature value in °C. And the height data analyzed from this are presented in Figure 5, where the analyzed height values in the stationary location float with changes in barometric pressure.

After the tag descended to −0.5 m, the height results calculated by Equation (4) were also approximately −0.5 m. This confirms that BMP sensors possess a reasonable level of accuracy for measuring height changes over short periods in indoor environments. However, after thirty minutes of observation, the height data drift estimated by a single BMP sensor reached up to nearly 1.5 m at its maximum. Consequently, the results from a single sensor demonstrate considerable uncertainty after several minutes. Due to the similarity in barometric drift trends observed with the two BMP sensors in the indoor environment, we propose a method using dual BMP sensors to more accurately measure the indoor tag height $H_{dual(t)}$ as illustrated in Equations (Equation 5) and (6): (5)$$\begin{array}{cc} H_{dual(t)} & =H_{Tag(t)}-H_{Anchor(t)} \end{array}$$(6)$$\begin{array}{cc} \; & =\left[{\left(\frac{P_{Anchor(t)}}{P_{Anchor(0)}}\right)}^{\frac{1}{5.257}}-{\left(\frac{P_{Tag(t)}^{\prime }}{P_{Anchor(0)}}\right)}^{\frac{1}{5.257}}\right]\cdot K_{temp}\;. \end{array}$$
where $P_{Anchor(0)}$ is the initial barometric pressure of Anchor (A) and serves as the reference barometric pressure. In the indoor environment, the temperatures of the tag and the anchor are similar, so the temperature parameter value is uniformly set to $K_{temp}$, which has a value of 44,330. The relative height was calculated as shown in red in Figure 5. After 30 min, the estimated tag height drifted by about 0.2 m, which is two-fifteenths of the error from a single sensor. This result has practical value for measuring tag height in indoor environments.

In the designed localization system, the filtered barometric pressure value from the tag is transmitted to the main controller via Wi-Fi. The time spent on Wi-Fi communication can impact the stability of the main process. Thus, this study leverages the dual-core processing capability of the controller (ESP32), dedicating one core specifically to handle Wi-Fi data transmission. Consequently, the data output of the localization system is accelerated and stabilized. Meanwhile, if the localization duration extends, the height estimated by the dual BMP sensors may still experience some drift. Therefore, the designed localization system includes a barometric pressure recalibration function. When the ranging part of the UWB sensor detects that the tag is very close to Anchor (A), the barometric pressure difference $P_{diff}$ is updated as follows:(7)$$\begin{array}{cc} \; & P_{diff}=P_{Anchor(t)}-P_{Tag(t)}\;. \end{array}$$

Meanwhile, the height estimation is further optimized in the designed 3D localization algorithm.

### 3.2. Distance Measurement Optimization for UWB Sensors

Patch-type UWB sensors, limited by their physical structure, may have ranging affected by orientation [33]. In this study, UWB sensors based on the new generation DW3000 chip (manufactured by Qorvo, Greensboro, NC, USA) are used for ranging. They also feature 2 dBi gain antennas to enhance the UWB sensors’ omnidirectional ranging capabilities. The ranging principle involves recording the time it takes to transmit and receive extremely short pulse signals, which is used to calculate the distance between two devices [22].

Similar to BMP sensors, UWB sensors are also subject to measurement noise. Noise data or sudden erroneous extreme values can compromise the accuracy of the localization system. Therefore, filters are applied to refine the raw data output from the UWB sensors, obtaining more stable distance measurements $L_{Filter}$. However, due to the hardware limitations of UWB sensors, filtered UWB measurements still show some errors compared to the actual distances. For example, the actual distance between two UWB sensors is 3 m, but the filtered output from the sensor devices is 3.18 m. Therefore, more measured and actual values need to be sampled to calibrate the UWB sensor [34,35].

Considering the range of indoor measurements, we sampled distance values at multiple fixed points from 0.1 m to 10 m. As illustrated in Figure 6, these measurement results were used to fit a calibration equation using Matlab’s Curve Fitting Toolbox, resulting in the following equation:(8)$$\begin{array}{c} L_{UWB(t)}=a\cdot L_{Filter(t)}+b\;, \end{array}$$
where the fitting parameter *a* is 0.9954, parameter *b* is −16.02, and the confidence bound for the fitting coefficients is 95%. When the filtered sensor measurement $L_{Filter(t)}$ is input in real time, the system outputs the calibrated value $L_{UWB(t)}$. The final distance estimate obtained after calibration is closer to the real distance.

## 4. Indoor 3D Localization Method

For 3D localization of the tag, a minimum of four reference anchors is typically required, with additional anchors used to enhance localization accuracy [36]. In this study, we use three UWB sensors at the same height as the localization anchor points to establish a geometric localization model for tag 3D localization. Furthermore, BMP sensors are utilized to determine the tag’s height relative to a reference plane, as well as enhancing the height estimation’s accuracy. The designed localization system is approached as in Algorithm 1.
**Algorithm 1** Indoor 3D localization method.**Input:** Distance values measured by UWB sensors: $L_{UWB_{A}\left(t\right)},L_{UWB_{B}\left(t\right)},L_{UWB_{C}\left(t\right)}$.
   Barometric pressure values measured by BMP sensors: $P_{Anchor(t)},P_{Tag(t)}$.
**Output:** The coordinates of the tag’s 3D indoor location: $\left(Tag_{x(t)},Tag_{y(t)},Tag_{z(t)}\right)$.
1:**if** $\left(L_{UWB_{A}\left(t\right)}<A\;\mathrm{predefined}\;\mathrm{close}\;\mathrm{distance}\;\mathrm{value}\;\mathrm{for}\;\mathrm{calibration}\right)$ **then**2:    $P_{diff}\leftarrow P_{Anchor(t)}-P_{Tag(t)}$3:**end if**4:$P_{Tag(t)}^{\prime }\leftarrow P_{Tag(t)}+P_{diff}$5:$H_{dual(t)}\leftarrow \mathrm{Dual}\;\mathrm{BMP}\;\mathrm{sensor}\;\mathrm{height}\;\mathrm{calculation}\;\mathrm{with}\;\mathrm{Equation}\;\left(6\right)$6:$P_{z(t)}\leftarrow H_{dual(t)}$7:${\bar{L}}_{UWB_{A}\left(t\right)},{\bar{L}}_{UWB_{B}\left(t\right)},{\bar{L}}_{UWB_{C}\left(t\right)}\leftarrow H_{dual(t)},L_{UWB_{A}\left(t\right)},L_{UWB_{B}\left(t\right)},L_{UWB_{C}\left(t\right)}\mathrm{with}\;\mathrm{Equation}\;\left(9\right)$8:$P_{x(t)},P_{y(t)}\leftarrow \mathrm{Geometric}\;\mathrm{Localization}\;\mathrm{Model}\left({\bar{L}}_{UWB_{A}\left(t\right)},{\bar{L}}_{UWB_{B}\left(t\right)},{\bar{L}}_{UWB_{C}\left(t\right)}\right)$9:$Tag_{x(t)},Tag_{y(t)},Tag_{z(t)}\leftarrow \mathrm{Kalman}\;\mathrm{Filter}\;\mathrm{Function}\left(P_{x(t)},P_{y(t)},P_{z(t)}\right)$10:**return** $\left(Tag_{x(t)},Tag_{y(t)},Tag_{z(t)}\right)$


### 4.1. Geometric Localization Model

Anchors for localization are distributed in an indoor space. The UWB sensors on the three anchors are adjusted to the same height $H_{Tripod}$ using tripods. In this horizontal plane, Anchor (A) is positioned midway between Anchor (B) and Anchor (C). As depicted in Figure 7, the geometry model is based on these three anchors. The tag location is denoted as point P. The locations of the three anchors are denoted as points A, B, C. The point O is the midpoint of line segment BC. Therefore, the localization system is defined with the x-axis in the direction of $\vec{\mathrm{OA}}$, the y-axis in the direction of $\vec{\mathrm{BC}}$, and the z-axis facing vertically upwards. The lines AP, BP, CP represent the true length of the tag from each anchor. The projection of the tag point P in the plane is the point K, and the coordinates $\left(P_{x},P_{y},P_{z}\right)$ of point P need to be computed.

The UWB sensors at each anchor point measure the distance ($L_{UWB_{A}\left(t\right)}$, $L_{UWB_{B}\left(t\right)}$, $L_{UWB_{C}\left(t\right)}$) from the tag point P in real time. At a given moment, these moments are denoted as $\hat{AP}$, $\hat{BP}$, and $\hat{CP}$, respectively. To calculate the tag’s coordinates $\left(P_{x(t)},P_{y(t)}\right)$ in the X-axis and Y-axis, it is necessary to project the UWB measurements in the real 3D environment to the anchors’ planes. Equation (Equation 9) employs the Pythagorean theorem to calculate the projected distance values from the UWB sensors, integrating the tag’s height information estimated by dual BMP sensors as the PK in the geometric model. The lengths obtained are ${\bar{L}}_{UWB_{A}\left(t\right)}$, ${\bar{L}}_{UWB_{B}\left(t\right)}$, ${\bar{L}}_{UWB_{C}\left(t\right)}$, corresponding to line segments $\hat{AK}$, $\hat{BK}$, and $\hat{CK}$, respectively:(9)$$\begin{array}{c} {\bar{L}}_{UWB(t)}=\sqrt{{L_{UWB(t)}}^{2}-{H_{dual(t)}}^{2}}\;. \end{array}$$

Within the plane of the anchors, circles are drawn with the anchor as the center and the measured projected distance as the radius, respectively. For instance, a circle is drawn with the Anchor C as the center and $\hat{CK}$ as the radius. Ideally, the three circles should intersect at point K. However, since the lengths $\hat{AP}$, $\hat{BP}$, and $\hat{CP}$ estimated by the UWB sensor are approximations, the circles might intersect at multiple points as depicted in Figure 8. The three intersections ${\bar{P}}_{BC}$, ${\bar{P}}_{AC}$, ${\bar{P}}_{BC}$ that are closest to each other are selected to form a triangle, and its centroid is used as the estimated point K. To determine the centroid’s position, each of the two circles needs to provide a vertex for the final triangle.

The geometric calculation starts by determining whether these circles intersect, based on the locations of points A, B, and C, and the lengths of $\hat{AP}$, $\hat{BP}$, and $\hat{CP}$. When each pair of circles has two intersection points, a total of six intersection points are generated. Additionally, due to the impact of UWB data noise, the measured results for $\hat{AP}$, $\hat{BP}$, and $\hat{CP}$ may result in the circles not intersecting. In this case, the vertex between the two circles needs to be redefined. Therefore, this study addresses several potential geometric scenarios separately. As an example, the intersection of circles *A* and *C* is divided into two cases for solving the intersection’s result. Then, the results of the other two sets of two circles are combined to find the triangle’s centroid.

Case 1: Two circles intersect at two points.

There are two intersections when two circles intersect. As shown in Figure 9, the intersection points $P_{1}$ and $P_{2}$ to be found are symmetrical about the line segment AC. The area of $▵ACP_{1}$ is calculated using Heron’s formula, and the lengths of $P_{1}G$ and CG are determined as shown in Equations (Equation 10)–(12).
(10)$$\begin{array}{cc} S_{▵ACP_{1}} & =\sqrt{f\cdot \left(f-AC\right)\cdot \left(f-AP_{1}\right)\cdot \left(f-CP_{1}\right)},\;\left(where,\;f=0.5\cdot \left(AC+AP_{1}+CP_{1}\right)\right)\;. \end{array}$$
(11)$$\begin{array}{cc} P_{1}G & =2\cdot S_{▵ACP_{1}}/AC\;, \end{array}$$
(12)$$\begin{array}{cc} CG & =\sqrt{{CP_{1}}^{2}-{P_{1}G}^{2}}\;, \end{array}$$

Then, the coordinates of point G can be obtained by the principle of similar triangles, i.e., by calculating the ratio of the lengths of CG to AC. Therefore, based on the point G, the coordinates of the points $P_{1}$ and $P_{2}$ in this plane are solved from the perpendicular relation and the length of $P_{1}G$. Then, the point $P_{2}$ inside the circle *B* is chosen as the coordinates of ${\bar{P}}_{AC}$. Using the same method, ${\bar{P}}_{BC}$ and ${\bar{P}}_{BC}$ are determined through geometric calculation.

Case 2: Two circles are tangent or non-intersecting.

As in Figure 10, the inclusion relationship between circle *A* and circle *C* is determined by the length relationship between $\hat{AP}$, $\hat{CP}$, and AC. When two circles are tangents, there is a shared tangent point that can be used as a vertex (${\bar{P}}_{AC}$) of the required triangle. And when two circles do not have an intersection point, it can be the case that the two circles do not contain each other, or that one circle contains the other. In this case, it is necessary to determine the point ${\bar{P}}_{AC}$ in these unsolved cases by the percentage of $\hat{AP}$ and $\hat{CP}$.

The cases where the two circles are tangents or non-intersecting can be calculated as Equations (Equation 13) and (14): (13)$$\begin{array}{cc} {\bar{P}}_{AC_{x}} & =\left\{\begin{array}{cc} A_{x}+\left(C_{x}-A_{x}\right)\cdot \left(\hat{AP}/\left(\hat{AP}+\hat{CP}\right)\right) & \left(\mathrm{if}\;AC\geq \left(\hat{AP}+\hat{CP}\right)\;\right)\\ A_{x}-\left(OA/AC\right)\cdot \left(\hat{AP}+AC+\hat{CP}\right)/2 & \left(\mathrm{if}\;\hat{AP}\geq \left(AC+\hat{CP}\right)\;\right)\\ C_{x}+\left(OA/AC\right)\cdot \left(\hat{AP}+AC+\hat{CP}\right)/2 & (\mathrm{if}\;\hat{CP}\geq \left(AC+\hat{AP}\right)\;)\;, \end{array}\right. \end{array}$$(14)$$\begin{array}{cc} {\bar{P}}_{AC_{y}} & =\left\{\begin{array}{cc} A_{y}+\left(C_{y}-A_{y}\right)\cdot \left(\hat{AP}/\left(\hat{AP}+\hat{CP}\right)\right) & \left(\mathrm{if}\;AC\geq \left(\hat{AP}+\hat{CP}\right)\;\right)\\ A_{y}+\left(OC/AC\right)\cdot \left(\hat{AP}+AC+\hat{CP}\right)/2 & \left(\mathrm{if}\;\hat{AP}\geq \left(AC+\hat{CP}\right)\;\right)\\ C_{y}-\left(OC/AC\right)\cdot \left(\hat{AP}+AC+\hat{CP}\right)/2 & (\mathrm{if}\;\hat{CP}\geq \left(AC+\hat{AP}\right)\;)\;, \end{array}\right. \end{array}$$
where $\left(A_{x},A_{y}\right)$ represents the coordinate of point A. $\left(C_{x},C_{y}\right)$ represents the coordinate of point C, and $\left(P_{AC_{x}},P_{AC_{y}}\right)$ is the coordinate of point ${\bar{P}}_{AC}$. The three scenarios in the piecewise function correspond to the following: the two circles do not encompass each other; circle *A* contains circle *C*; and circle *C* contains circle *A*. Here, when the judgment in the equation is an equal sign, it represents scenarios where the two circles are tangents.

Similarly, the results of geometrically resolving circles *A* and *B* and circles *B* and *C*, respectively, result in a total of three intersections: ${\bar{P}}_{BC}$, ${\bar{P}}_{AC}$, and ${\bar{P}}_{BC}$. Figure 11 shows a possible scenario where circles *A* and *C* do not intersect but circle *B* intersects circles *A* and *C*, respectively. In addition, for any other possible scenarios, it is possible to solve for three sets of two circles based on Case 1 and Case 2, respectively, thus forming a triangle that can be used to solve for the centroid *K*.

Finally, by the centroid method, the location $\left(P_{x},P_{y}\right)$ of the point P projected on the anchors’ plane is estimated. Then, the tag’s height $H_{dual(t)}$ estimated by the dual BMP sensor is used again. The tag’s 3D coordinates are denoted as:(15)$$\begin{array}{c} \left\{\begin{array}{cc} P_{x} & =\frac{1}{3}\left({\bar{P}}_{AB_{x}}+{\bar{P}}_{AC_{x}}+{\bar{P}}_{BC_{x}}\right)\\ P_{y} & =\frac{1}{3}\left({\bar{P}}_{AB_{y}}+{\bar{P}}_{AC_{y}}+{\bar{P}}_{BC_{y}}\right)\\ P_{z} & =H_{dual}\;. \end{array}\right. \end{array}$$

### 4.2. Optimization of Location Estimation Based on Kalman Filtering

Due to the fluctuations in the data measured by the sensors, the tag’s location calculated using the geometric model may deviate from the actual value. The Kalman filtering algorithm is an optimization estimation algorithm that is very effective in dealing with linear dynamic systems containing Gaussian noise. To optimize the location estimation, this study employs the Kalman filtering method to predict and update the optimal coordinates of the tag.

First, the time node information of the running loop in the localization system can be used to estimate the tag’s velocity $v_{\left(t-1\right)}$ at the last moment (t−1):(16)$$\begin{array}{cc} v_{\left(t-1\right)} & =\left[\begin{array}{c} v_{x(t-1)}\\ v_{y(t-1)}\\ v_{z(t-1)} \end{array}\right]=\left[\begin{array}{c} \Delta P_{x(t-1)}\\ \Delta P_{y(t-1)}\\ \Delta P_{z(t-1)} \end{array}\right]\cdot \frac{1}{\Delta t_{\left(t-1\right)}}\;, \end{array}$$
where the tag’s velocity $v_{\left(t-1\right)}$ in the three dimensions consists of the components [$v_{x(t-1)}$, $v_{y(t-1)}$, $v_{z(t-1)}$]′. The tag’s height $P_{z(t-1)}$ is equal to the height estimated by the dual BMP sensors $H_{dual(t-1)}$. Δt denotes the change, with the change in locations represented by [$\Delta P_{x(t-1)}$, $\Delta P_{y(t-1)}$, $\Delta P_{z(t-1)}$]′.

Assume that the labels have similar velocities in a very short period of time. Then, the predicted location $X_{\left(t|t-1\right)}$ at the current moment is as follows:(17)$$\begin{array}{cc} U_{\left(t\right)} & =v_{\left(t-1\right)}\cdot \Delta t_{\left(t\right)}\;, \end{array}$$(18)$$\begin{array}{cc} X_{\left(t|t-1\right)} & =X_{\left(t-1|t-1\right)}+U_{\left(t\right)}\;. \end{array}$$
where the subscript (t|t−1) denotes the estimation of the (t) moment based on the (t−1) moment. The $X_{\left(t-1|t-1\right)}$ is the optimal solution for the location estimated at the previous moment, [$P_{x(t-1)}$, $P_{y(t-1)}$, $P_{z(t-1)}$]′. And the control input $U_{\left(t\right)}$ is the predicted location increment.

Then, the update part of the Kalman filter for the localization data is as follows:(19)$$\begin{array}{cc} P_{\left(t|t-1\right)} & =A\cdot P_{\left(t-1|t-1\right)}\cdot A^{T}+Q\;, \end{array}$$(20)$$\begin{array}{cc} K_{\left(t\right)} & =P_{\left(t|t-1\right)}H^{T}{\left(HP_{\left(t|t-1\right)}H^{T}+R\right)}^{-1}\;, \end{array}$$(21)$$\begin{array}{cc} Z_{\left(t\right)} & ={\left[P_{x(t)},P_{y(t)},P_{z(t)}\right]}^{\prime }, \end{array}$$(22)$$\begin{array}{cc} {\hat{X}}_{\left(t|t\right)} & ={\hat{X}}_{\left(t|t-1\right)}+K_{\left(t\right)}\cdot \left(Z_{\left(t\right)}-X_{\left(t|t-1\right)}\right)\;, \end{array}$$(23)$$\begin{array}{cc} P_{\left(t|t\right)} & =\left(I-K_{\left(t\right)}\right)\cdot P_{\left(t|t-1\right)}\;. \end{array}$$

Herein, P represents the prediction error covariance, $K_{\left(t\right)}$ is the Kalman gain, and ${\hat{X}}_{\left(t|t\right)}$ is the state estimation. *A* is the state transition matrix, *H* is the observation matrix, and *I* is a 3×3 identity matrix. $Z_{\left(t\right)}$ is the actual measurement at the current moment, and its data are the tag’s coordinate data calculated by the geometric localization model in Equation (Equation 15). Referring to the tests of UWB and BMP sensors in Section 3, the process noise covariance matrix *Q* in the Kalman filter is set to $diag(\left[1\times 10^{-4},1\times 10^{-4},1\times 10^{-3}\right])$, and the measurement noise covariance matrix *R* is set to $diag(\left[4\times 10^{-4},4\times 10^{-4},5\times 10^{-3}\right])$. Finally, the positioning system’s output at the current moment is ${\hat{X}}_{\left(t|t\right)}$, i.e., the estimated location of the tag [$Tag_{x(t)}$, $Tag_{y(t)}$, $Tag_{z(t)}$]′.

In addition, at the start of the localization system operation, the Kalman filter cannot function accurately immediately due to the lack of initial measurement data for prediction. Therefore, at the first moment, the filter does not perform computations, and its output is directly set to the input positioning data. The initial error covariance matrix is set as a diagonal matrix diag[1, 1, 1], and from the second moment onwards, the filter begins to operate formally.

This section introduced the proposed indoor 3D localization method utilizing sensor fusion. UWB sensors at three anchors calculated distances to the tag, while dual BMP sensors provided height estimates, enabling the mapping of UWB sensors’ range measurements onto a 2D plane. The geometric localization model, based on these measurements, integrated various relationships and compensated for errors. The tag’s location on the anchor’s plane was determined using the centroid method, and Kalman filtering was applied to enhance the accuracy of the 3D location estimates. The scheme will be validated experimentally in the following section.

## 5. Experiments and Evaluations

### 5.1. Experimental Setup

The experimental validations were located in indoor environments. One setting was in a laboratory equipped with a motion capture system, and the other was a hall with more space. The experimental setup for the two scenarios is shown in Figure 12. The UWB sensors in the individual anchors were adjusted to the same height $H_{Tripod}$ by means of a tripod, and the parameters of the anchors’ positions are listed in Table 2.

In the laboratory environment, a motion capture system (produced by OptiTrack) was used to capture the tag’s reference positions, with the marker fixed on the tag. This system covers a measurable area of 3 m (length) × 2.5 m (width) × 2 m (height) with millimeter accuracy. Results from the motion capture system serve as reference locations for the tag, used to evaluate the localization system’s accuracy. Additionally, to verify the maximum locatable range of the proposed system, a static tag localization experiment was carried out in a spacious hall setting.

Moreover, Figure 12 shows the base coordinate systems of both the proposed localization system and the motion capture system. The base coordinate system of the localization system {O} is as described in Section 4.1’s geometric localization model. The base coordinate system of the motion capture system {MCS} is established through a device containing three MCS markers, positioned directly below {O} and parallel to the ground. As the frames of the two systems reside in different coordinate systems, the registration of two frames is required to compare data. Each frame of the motion capture system is transformed into the proposed localization system as follows:(24)$$\begin{array}{c} P_{M,i}^{\prime }=R_{\mathrm{MCS}}^{O}\cdot P_{M,i}+d_{\mathrm{MCS}}^{O}=\left[\begin{array}{ccc} 0 & 0 & 1\\ 1 & 0 & 0\\ 0 & 1 & 0 \end{array}\right]\cdot \left[\begin{array}{c} x_{P_{M},i}\\ y_{P_{M},i}\\ z_{P_{M},i} \end{array}\right]+\left[\begin{array}{c} 0\\ 0\\ -H_{Tripod} \end{array}\right]=\left[\begin{array}{c} z_{P_{M},i}\\ x_{P_{M},i}\\ y_{P_{M},i}-H_{Tripod} \end{array}\right]\;. \end{array}$$
where $P_{M,i}$ is the i-th reference location data in the {MCS} coordinate system. It is transformed to the coordinate location $P_{M,i}^{\prime }$ in the {O} coordinate system by a rotation matrix $R_{\mathrm{MCS}}^{O}$ and a translation matrix d. The specific values of the two matrices are obtained in accordance with the location information in the schematic. The result of the transformation is $\left(z_{P_{M},i},\;x_{P_{M},i},\;y_{P_{M},i}-H_{Tripod}\right)$. It should be declared that there is a registration error in the two-frame registration process. The measurement point $P_{O}$ of the localization system is the tag’s UWB antenna, and the measurement point $P_{M}$ of the motion capture system is the MCS marker. Due to the deviation of about 2 cm between the measurement points, the attitude change when the tag moves may slightly affect the registration accuracy.

In addition, the Root Mean Square Error (RMSE) is a common measure of the difference between the estimated and reference values as defined in Equation (Equation 25). Here, $Err_{i}$ denotes the Euclidean distance between the i-th estimated location and its corresponding reference location. A smaller RMSE value indicates a lesser difference between the estimated and the reference values, signifying higher localization accuracy:(25)$$\begin{array}{c} \mathrm{RMSE}=\sqrt{\frac{1}{n}\sum_{i=1}^{n}{\left(Err_{i}\right)}^{2}}\;. \end{array}$$

After registering the frames of the localization system to the frames of the motion capture system, the specific definition of $Err_{i}$ for different error evaluation objects is as follows:Errors on different coordinate axes ($Err_{x,i}$, $Err_{y,i}$, $Err_{z,i}$):
(26)$$\begin{array}{cc} Err_{x,i} & =x_{ref,i}-x_{est,i}\;, \end{array}$$
(27)$$\begin{array}{cc} Err_{y,i} & =y_{ref,i}-y_{est,i}\;, \end{array}$$
(28)$$\begin{array}{cc} Err_{z,i} & =z_{ref,i}-z_{est,i}\;. \end{array}$$Two-dimensional localization error on the X-Y plane ($Err_{2D,i}$):
(29)$$\begin{array}{c} Err_{2D,i}=\sqrt{{\left(Err_{x,i}\right)}^{2}+{\left(Err_{y,i}\right)}^{2}}\;. \end{array}$$Three-dimensional localization error across the X-Y-Z axes ($Err_{3D,i}$):
(30)$$\begin{array}{c} Err_{3D,i}=\sqrt{{\left(Err_{x,i}\right)}^{2}+{\left(Err_{y,i}\right)}^{2}+{\left(Err_{z,i}\right)}^{2}}\;. \end{array}$$
where $x_{ref,i}$, $y_{ref,i}$, $z_{ref,i}$ are the tag’s reference location data collected by the motion capture system. $x_{est,i}$, $y_{est,i}$, and $z_{est,i}$ are the estimated location data of the proposed localization system.

### 5.2. Indoor 3D Localization Experiment

The experiment aims to verify the accuracy of the indoor 3D localization system, particularly the accuracy of the height estimation. For this purpose, the tag’s moving reference trajectory was set as two rectangles (1.4 m × 1.6 m) with different heights. The reference locations of the tag were recorded by the high-precision motion capture system.

The tag’s trajectory, as shown in Figure 13, began at the point labeled ’Start’, moved around a horizontal rectangle, then vertically up about 0.5 m. After completing another horizontal loop around the rectangle, it concluded at the point labeled ’End’. The tag’s reference trajectory is depicted as the ‘Reference’ in the figure, with the geometric localization results shown in blue circles. The final Kalman filtering results produced by the localization system are indicated by the red line. The system’s output rate was 37 Hz, processing nearly 2600 data sets in 70.5 s.

To better differentiate the accuracy of localization in each dimension, Figure 14 displays the comparison of the tag’s positions on the X-axis, Y-axis, and Z-axis. The gray background in the figure indicates the phase where the tag was ascending, flanked by rectangular path sections. It is evident that the system’s localization results closely align with the reference trajectory. On the X and Y axes, there are relatively larger errors at the corners of the rectangular path. The optimization by the Kalman filter significantly enhances the geometric localization results.

The analysis of the results from the indoor 3D localization experiment is presented in Table 3. For the final output data of the localization system, the RMSE was about 0.041 m for the X-axis and Y-axis, and 0.028 m for the Z-axis. When the errors across different axes were superimposed, the 2D and 3D localization errors were also increased. Additionally, the cumulative distribution function (CDF) shown in Figure 15 demonstrates the specific error distribution for the 3D localization results.

The accuracy of the tag’s height estimation is crucial for 3D localization. In the Z-axis part, the results of the height error estimated by the dual barometric pressure sensors are as shown in Figure 16. Before filtering, the height error of the geometric localization results was about ±0.1 m. After filtering, the error was reduced to approximately ±0.05 m.

### 5.3. Locatable Range Verification Experiment

To verify the maximum locatable range of the proposed system, we conducted experiments in a more spacious hall. The three anchors were located in the same planar position as the experimental setup, and the height of the anchors $H_{Tripod}$ was set to 1.3 m. As in Figure 17, the stationary reference points for the test were set at the four vertices of a 6 m (length) × 6 m (width) × 2 m (height) sized cube. Based on the coordinate system of the localization system, the positions of these four reference points were Point ① (−3,−3,−1.1), Point ② (3,3,−1.1), Point ③ (3,−3,0.9), and Point ④ (−3,3,0.9). For each reference point, the localization system sampled 380 localization data (about 10 s) separately. In Figure 17, the geometric localization results are shown in blue points, and the Kalman filtering results are shown in red points.

Table 4 presents the RMSE comparisons of 2D and 3D errors for four points, calculated against their respective reference points. And Figure 18 shows a detailed distribution of the 3D localization error results, where the blue boxes are geometric localization results, and the purple boxes are Kalman filtering results.

Analysis data and boxplots indicate that the Kalman filtering optimized the geometric localization results for four points. Notably, the error variability range narrowed, the median error decreased, and the average 3D localization RMSE improved from 0.0860 m to 0.0740 m. These results confirm the system’s locatability within a 6 m (length) × 6 m (width) × 2 m (height) range.

## 6. Discussion

In this section, we compare the proposed system with our previous study and other related studies, respectively.

### 6.1. Comparison with Our Previous Study

This study successfully implemented indoor 3D localization, with the proposed system being evaluated through two experiments. The first experiment displayed the localization trajectories of dynamic tags within a limited range, while the second demonstrated localization estimates of static tags over extended distances.

The 2D RMSE results for both experiments were nearly identical, approximately 0.058 m. For the 3D experimental results, the RMSE at longer distances was 0.074 m, which was an increase of 0.01 m compared to the experimental results obtained near the anchor.

Compared to our previous study [20], this research developed an enhanced indoor 3D localization system, with significant upgrades and updates to both hardware and software:In terms of hardware, the UWB sensor was upgraded from a patch type to an antenna type. To account for the propagation direction and range of the antenna signal, a short antenna with 2 dBi gain was selected to enhance the indoor omnidirectional ranging capability for the UWB sensors.Additionally, the main controller was upgraded from ESP8266 to the dual-core ESP32 chip. By rationally allocating tasks through the software, one core was specifically used for Wi-Fi data reading, thus making the data output of the localization system faster and more stable. The localization output rate reached 37 Hz, which was a nine-fold increase.The dispersed arrangement of the anchors allowed the locatable area of the new localization framework to expand. The verified locatable range of the system was 6 m (length) × 6 m (width) × 2 m (height), which was approximately three times larger compared to the previous localization device.In particular, the RMSE of the 3D localization system reached 0.074 m, which improved the localization accuracy by 40.7% compared to our previous study.

### 6.2. Comparison with Other Related Studies

Although a high-precision motion capture system was employed during the experimental validation, it uses fixed motion capture cameras that are not easily moved and are expensive. Our localization system features quick anchor setup, and the estimated tag trajectories closely match the reference values, offering more economic value in some application scenarios.

In the height estimation, localization using three UWB sensors makes it difficult to distinguish between positive and negative tag heights. A study on indoor 3D drone localization increased the accuracy of height estimation by adding a UWB sensor at a different height [13]. However, this study reports that an error of approximately 0.2 m still exists in the height estimation. Moreover, as bandwidth utilization increases, a lower sensor data output rate (5 Hz) significantly impacts the accuracy of the localization system. Ma’s research designed a 3D localization system for indoor mobile robots using four UWB anchors at different heights [31], compensating for the signal interference from patch-type UWB sensors, with decimeter-level accuracy meeting the application requirements. In areas with small height differences, a single BMP sensor shows insignificant pressure variations [6], and the accuracy is also greatly reduced by the effect of barometric drift. However, our proposed method using dual BMP sensors excels in height estimation, achieving an accuracy of ±0.05 m.

Table 5 and Table 6 compare the localization performance of more different studies by RMSE results. In 2D planar localization, our approach achieves over twice the accuracy compared to methods that utilize additional UWB sensors [13,37,38]. Zigbee-based localization methods are slightly less precise [39], and LiDAR faces challenges with repetitive localization accuracy [40].

In indoor 3D localization, a substantial number of sensors are required to support the localization in larger buildings, and significant barometric changes can estimate floor levels [6]. For the size of the room environments, an indoor localization method that integrates SLAM and UWB technologies demonstrate notable accuracy [41]. Yoon et al. developed a system that combines IMU and UWB, achieving high-accuracy localization in smaller areas for entertainment scenarios [42]. This study has achieved a 60% performance improvement over similar research with comparable measurement scopes as referenced in [13]. While the localization accuracy provided by four anchors at varying heights meets the needs of the intended scenarios [31], inaccuracies in any UWB range measurement can significantly increase the error in height estimation. To tackle this issue, our system incorporates dual BMP sensors to improve the precision of the height estimation. By utilizing geometric localization models and Kalman filters, the system’s indoor 3D localization RMSE is optimized to 0.074 m.

However, the system we propose is not without flaws. Continuous thick obstructions between UWB sensors can impact the accuracy of some distance measurements, leading to reduced overall localization accuracy. The height estimation of indoor dual BMP sensors has been validated. But the relative barometric pressure measurements can be unstable near air conditioning vents, which are independent of the overall indoor barometric pressure. Meanwhile, employing additional reference anchors will furnish the localization system with more measurement data. However, more non-linear factors must be considered in real-world localization applications. We plan to further optimize and explore these aspects in subsequent studies. Furthermore, limited by the size of the battery, future improvements could involve using smaller lithium batteries and integrating all components on the same PCB to further reduce the size of the tag. Alternatively, a smartwatch that integrates UWB sensors, BMP sensors, and Wi-Fi transceivers could serve as an alternative hardware for the tag.

## 7. Conclusions

This study developed an enhanced indoor 3D localization system utilizing UWB and BMP sensors. The system features dispersed anchors as reference points within an indoor environment. The anchors were set on tripods and could be easily arranged in new environments. Filters reduced measurement noise for both types of sensors. BMP sensors measured barometric pressure at various indoor heights. The barometric pressure value at the tag was sent to the main controller through a Wi-Fi enabled microcontroller. The tag’s relative height was estimated by comparing it with the barometric pressure value at the anchor point, and the error of the estimated height result was about ±5 cm. This height value is also used to project the UWB sensor’s measurements onto the anchor’s plane, which helps reduce errors in 2D localization. UWB sensors at the three anchors calculated distances to the tag. The established geometric localization model considered various geometric relations and compensated for potential errors. The tag’s projection location on the anchor’s plane was determined using the centroid method. Finally, Kalman filtering optimized the location estimation.

We validated the localization performance and locatable range of the proposed system through indoor 3D localization experiments. The system has fast and stable output. In particular, the height estimation scheme with dual barometers estimated the height results with an accuracy of about ±0.05 m. The RMSE of the 2D localization reached 0.0585 m, and the RMSE of the 3D localization reached 0.0740 m. Compared to indoor localization systems in similar environments, our system has a larger measurable range and higher localization accuracy.

In future research, we plan to use more anchors to provide measurement data for the indoor 3D localization system and consider more non-linear factors to optimize the system’s localization performance, with specific application scenarios such as indoor drones’ localization.

## Figures and Tables

**Figure 1 sensors-24-03341-f001:**
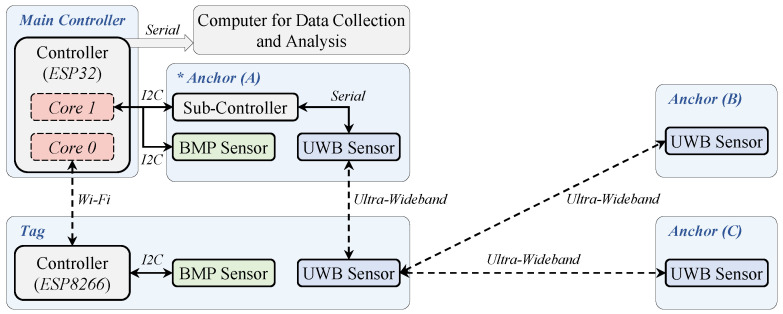
The hardware system framework.

**Figure 2 sensors-24-03341-f002:**
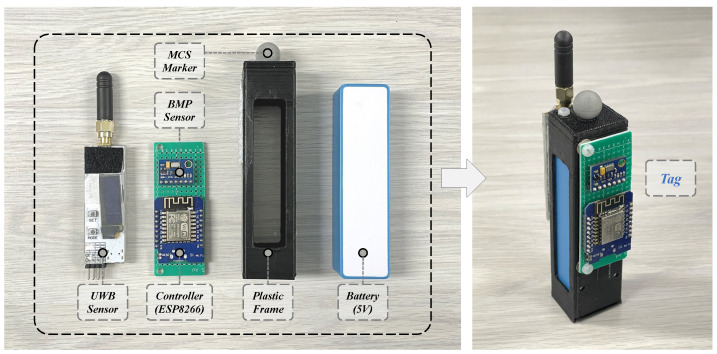
The tag’s hardware design.

**Figure 3 sensors-24-03341-f003:**
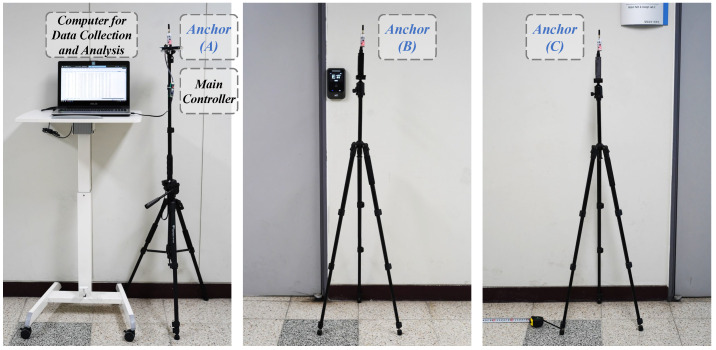
Hardware of the localization system’s anchors.

**Figure 4 sensors-24-03341-f004:**
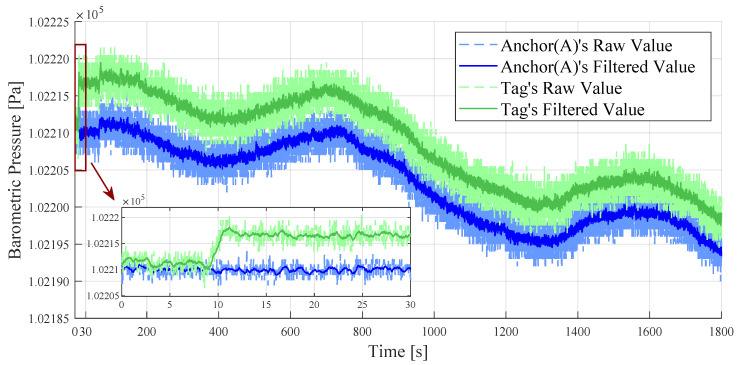
Barometric pressure values measured by BMP sensors and filtering results in 30 min.

**Figure 5 sensors-24-03341-f005:**
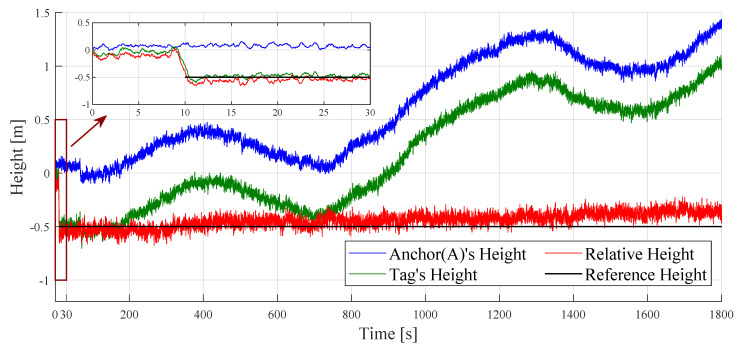
Estimated height values and relative height values based on dual BMP sensors in 30 min.

**Figure 6 sensors-24-03341-f006:**
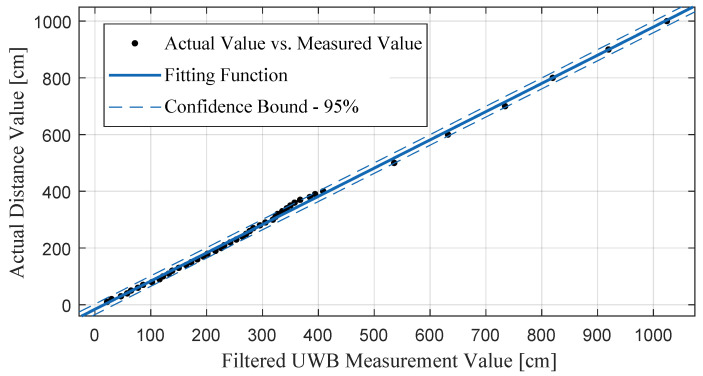
The calibration equation fitted based on the sampled measurement values and actual distance values.

**Figure 7 sensors-24-03341-f007:**
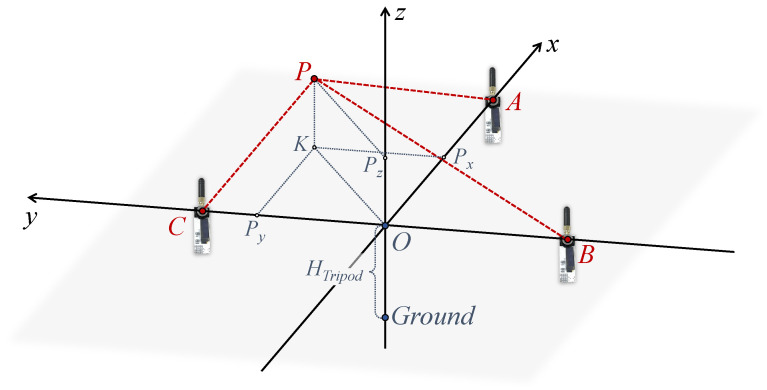
Geometric model based on the three anchors.

**Figure 8 sensors-24-03341-f008:**
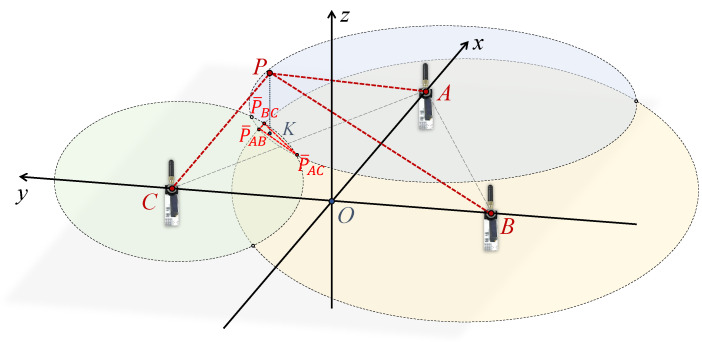
Geometric localization model.

**Figure 9 sensors-24-03341-f009:**
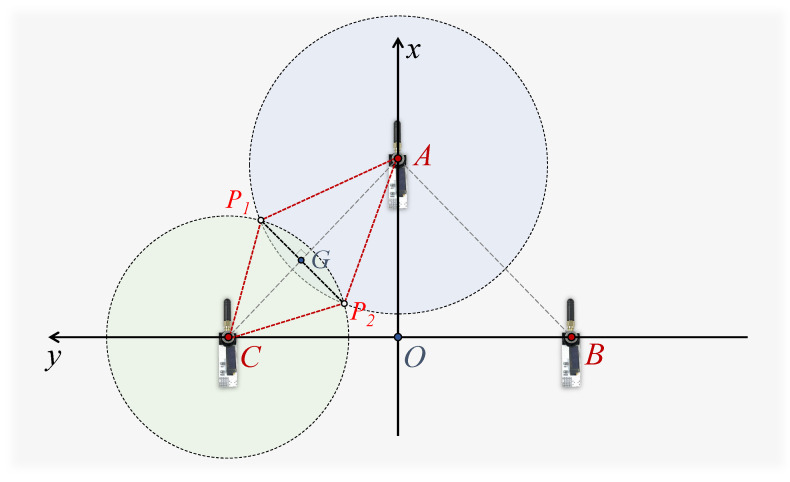
Geometric localization model of circle *A* and circle *C* intersecting at two points.

**Figure 10 sensors-24-03341-f010:**
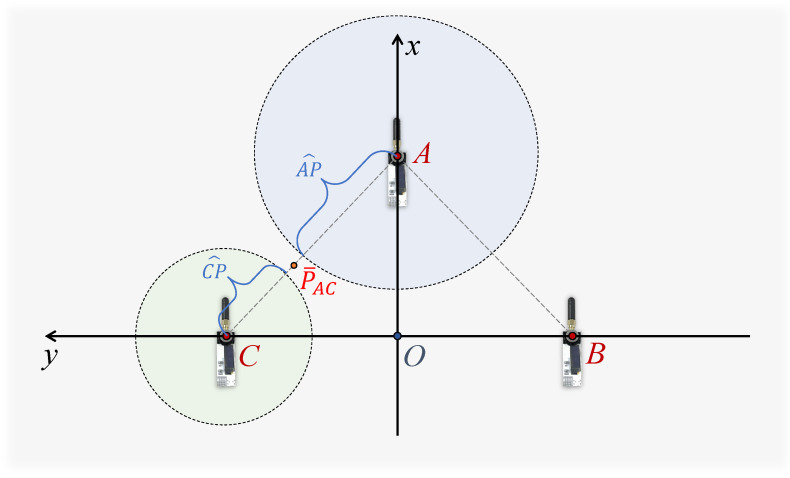
Geometric localization model of circle *A* and circle *C* without intersections.

**Figure 11 sensors-24-03341-f011:**
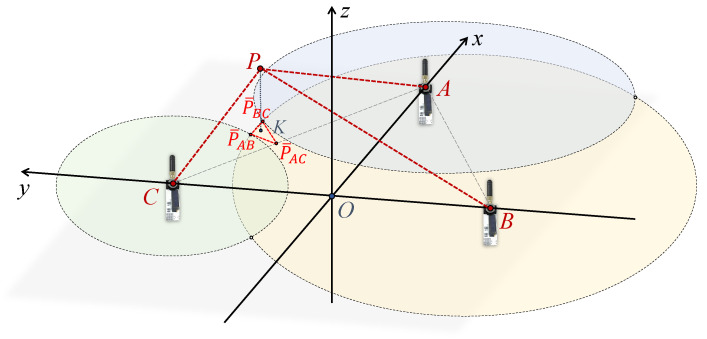
A possible scenario of the geometric localization model.

**Figure 12 sensors-24-03341-f012:**
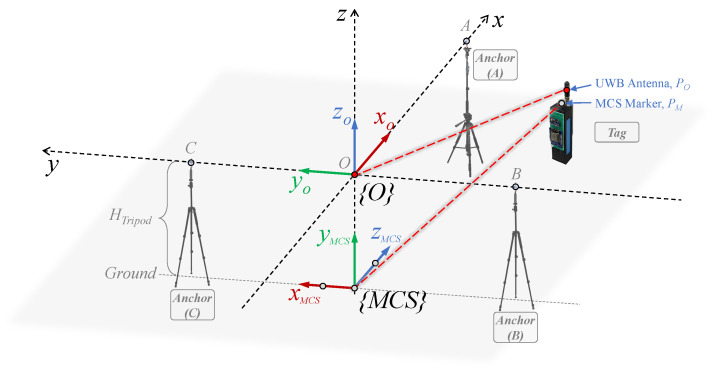
The schematic of the experimental setup.

**Figure 13 sensors-24-03341-f013:**
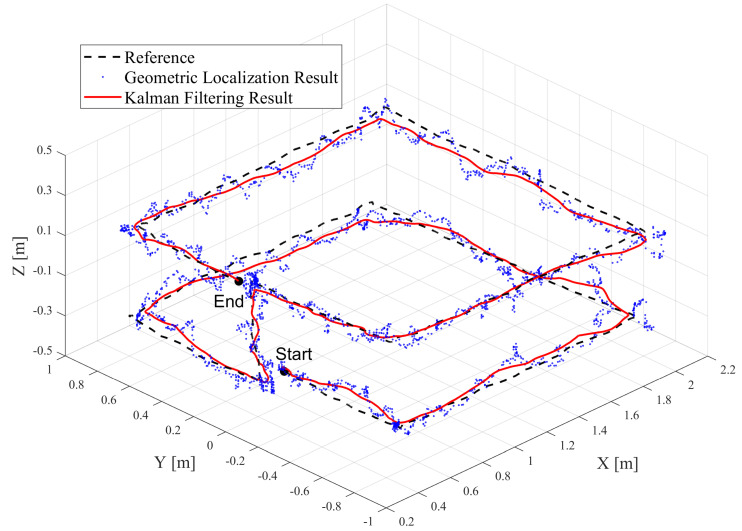
Results of the tag’s trajectories in the indoor 3D localization experiment.

**Figure 14 sensors-24-03341-f014:**
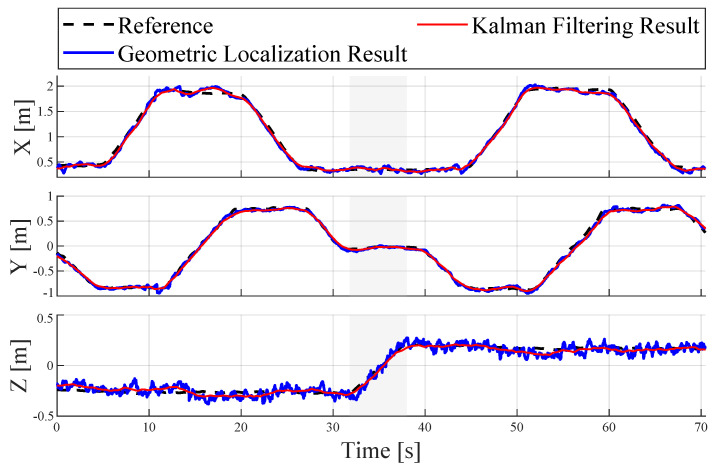
Comparison of the tag’s coordinates in three dimensions.

**Figure 15 sensors-24-03341-f015:**
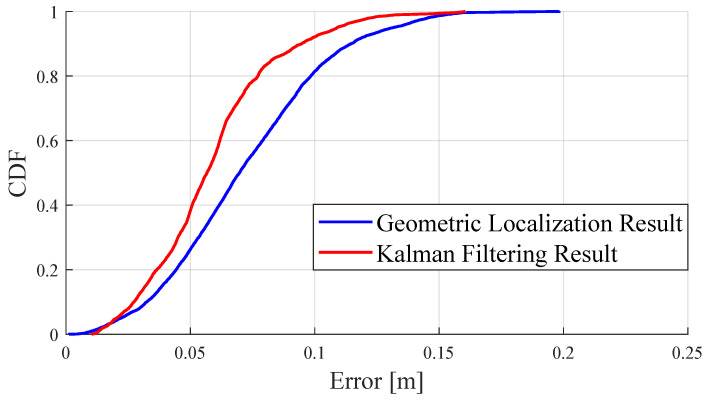
CDF results for the 3D localization errors in the localization experiment.

**Figure 16 sensors-24-03341-f016:**
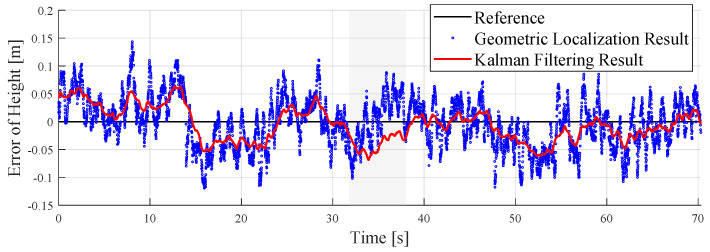
Results of the height errors in the localization experiment.

**Figure 17 sensors-24-03341-f017:**
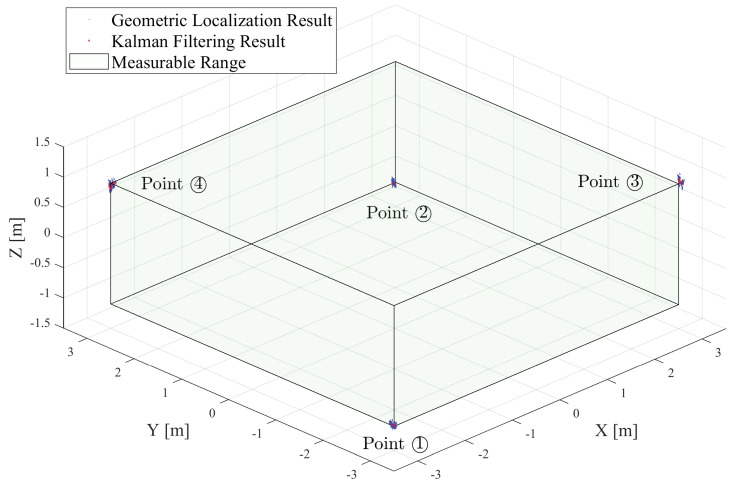
Results of the tag’s locations in the locatable range validation experiment.

**Figure 18 sensors-24-03341-f018:**
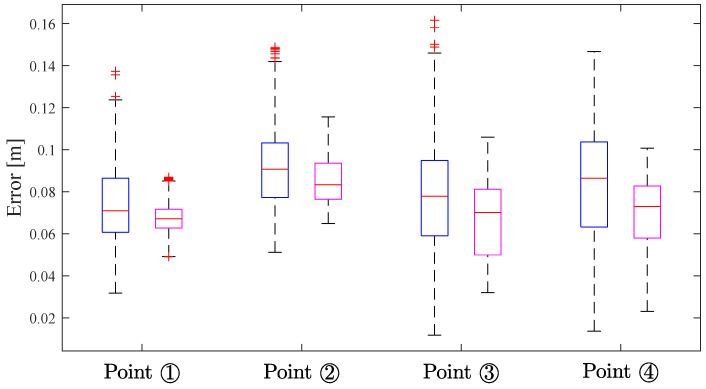
Boxplot results for the 3D localization errors in the locatable range validation experiment.

**Table 1 sensors-24-03341-t001:** The parameters of the devices used in the hardware system.

Device	Microchip	Board Model	Communication Mode
Controller (ESP32)	ESP32	Arduino Nano ESP32	Serial, I2C, Wi-Fi
Controller (ESP8266)	ESP8266	ESP8266 D1 Mini	I2C, Wi-Fi
Sub-Controller	ATmega2560	Arduino Mega	Serial, I2C
BMP Sensor	MS5611	GY-63	I2C
UWB Sensor	DW3000	D-DWM-PG3.9	Serial, UWB

**Table 2 sensors-24-03341-t002:** The parameters of the experimental setup.

$H_{\mathrm{Tripod}}$	OA	OB	OC
1.5 m	2.8 m	2.4 m	2.4 m

**Table 3 sensors-24-03341-t003:** Analysis of the indoor 3D localization experimental results.

Coordinate Axis	Localization Result	Maximum Error [m]	* RMSE [m]
X	Geometric	0.1524	0.0516
	Filtering	0.1137	0.0415
Y	Geometric	0.1374	0.0402
	Filtering	0.1403	0.0392
Z	Geometric	0.1434	0.0451
	Filtering	0.0616	0.0282
X-Y (2D)	Geometric	0.1781	0.0649
	Filtering	0.1602	0.0578
X-Y-Z (3D)	Geometric	0.1983	0.0790
	Filtering	0.1604	0.0643

* RMSE: Root Mean Square Error.

**Table 4 sensors-24-03341-t004:** Analysis of the locatable range validation experimental results.

Point	Localization Methods	RMSE (2D) [m]	RMSE (3D) [m]
①	Geometric	0.0575	0.0777
	Filtering	0.0541	0.0688
②	Geometric	0.0797	0.0928
	Filtering	0.0778	0.0856
③	Geometric	0.0599	0.0835
	Filtering	0.0592	0.0696
④	Geometric	0.0626	0.0900
	Filtering	0.0430	0.0720
Average Value	Geometric	0.0649	0.0860
	Filtering	0.0585	0.0740

**Table 5 sensors-24-03341-t005:** Comparison of 2D localization accuracy of different methods.

No.	Reference	Sensing Technology				Range	RMSE
		UWB	BMP	LiDAR	Zigbee	* L × W [m]	[m]
1	Proposed	✔	✔			6×6	0.058
2	[13]	✔	✔			5×5	0.158
3	[37]	✔				14×12	0.100
4	[38]	✔				15×10	0.560
5	[39]			✔		20×20	0.350
6	[40]				✔	4×4	0.636

* L × W: Length × Width.

**Table 6 sensors-24-03341-t006:** Comparison of 3D localization accuracy of different methods.

No.	Reference	Sensing Technology				Range	RMSE
		UWB	BMP	SLAM	IMU	* L × W × H [m]	[m]
1	Proposed	✔	✔			6×6×2	0.074
2	[6]	✔	✔			20×20×36	0.144
3	[13]	✔	✔			5×5×5	0.185
4	[31]	✔				5×5×3	0.145
5	[41]	✔		✔		6×6×4.5	0.139
6	[42]	✔			✔	2×2×1	0.130

* L × W × H: Length × Width × Height.

## Data Availability

Data are contained within the article.

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
