# Peer review of "An Enhanced Indoor Three-Dimensional Localization System with Sensor Fusion Based on Ultra-Wideband Ranging and Dual Barometer Altimetryâ€"

_sensors, 2024, doi:10.3390/s24113341_

Round 1
Reviewer 1 Report
Comments and Suggestions for Authors
Please see the review in the attached file.

Comments on the Quality of English LanguageExamples of hard-to-read sentences:'
'The results indicate a significant reduction in localization error, with height accuracy results was approximately ...' - lines #9-10
'Subsequently, the barometric pressure drift trends of the two stationary locations prove to be almost identical.' - lines #157-158
'Despite this, the drift trends observed in the two BMP sensors within the indoor environment were similarity.' - lines #172-174
Reviewer 2 Report
Comments and Suggestions for Authors
This paper reports a novel Enhanced Indoor Three-Dimensional Localization System. The Sensor Fusion based on Ultra-wideband Ranging and Dual Barometer Altimetry, are the novelty in this paper.
The idea is interesting, and the authors successfully board the problem challenges around building the hardware system design, including mathematical equations and an analysis of the simulation results.
The extension version is correctly and including more details and research.
The authors need to address the next concerns in order to present the paper, as described below:
In section IV, the authors can be summarized the Indoor 3D Localization section.
Reviewer 3 Report
Comments and Suggestions for Authors
This paper proposes an indoor localization method enhanced by BMP sensors for the UWB localization system. The performance is assessed by comparing it with a high-precision motion capture system. However, this comparison alone does not adequately validate the superiority of the proposed system, and certain assertions lack persuasiveness.
1. In the Abstract, it is stated that "significant improvements in data processing speed and stability" and a "significant reduction in localization error". Nevertheless, this paper does not provide a comparative analysis of the proposed method with any existing UWB localization methods.
2. Equation (8) duplicates Equation (2).
3. The notation "a" and "b" in (9) should follow the same format as in Equation (9).
4. A notable limitation is that the proposed method necessitates all anchors to be at the same height, which should be clarified towards the end of Section 1.
5. What does "P’" represent in line 256?
6. If circles A, B, and C do not intersect, is there a method to find a solution as depicted in Figure 8? It is recommended to utilize Case 1 and Case 2 to identify the scenarios discussed in Section 4.1.
7. Can the proposed method be extended to scenarios with more anchors?
8. The notation usage should be enhanced. Matrices and vectors should be represented in bold.
9. The measurement equation appears to be missing in Section 4.2. Furthermore, how does one initiate the Kalman filter?
10. Regarding the statement "a motion capture system (produced by OptiTrack) is used to capture the tag’s actual positions," please note that the actual positions are unknown. The positions provided by OptiTrack are still measured values.
11. The simulation does not compare with any state-of-the-art work, thus the necessity of including BMP sensors is not convincing. UWB can also achieve 3D localization. If there are four UWB nodes not in the same plane, the accuracy should be sufficient, including height.
12. How was the maximum measurable range of the system concluded from "the designed indoor 3D localization system’s maximum measurable range is approximately 8 m (length) × 8 m (width) × 3 m (height)"?
Comments on the Quality of English LanguageThe writing can be improved.
Reviewer 4 Report
Comments and Suggestions for Authors
This paper propose an accurate 3D localization scheme by using barometer-based height estimation and UWB-based position estimation. It introduces some reference barometers to calibrate the environmental air pressure in real time. Meanwhile, a geometry method is proposed to estimate target position based on the range obtained from UWB. By fusing through a Kalman filter, the accurate 3D location is then computed.
The height estimation with real time barometer calibration is practically useful. The authors also provide a detailed implementation description. Beside, this paper is well-written and easy to follow. Figures are illustrative and very helpful to understand.
There is some points that authors could further polish the paper.
- In Section 3.2, the authors propose a linear relationship between the filtered results and UWB distance estimation. I don't fully understand the rationale here. Normally, the expectation of the UWB distance estimation and the filtered result should be both equal to the true distance. However, if we take expectation of the linear equation, these two expectation values differs. It's counterintuitive and needs a justification.
- In the experiment part, the comparison schemes are missing. Though the author compare with other schemes by citing their reported results, they cannot be directly compared because of the inconsistent test settings. Meanwhile, the proposed system should be studied in depth. For example, the system performance under different number of barometer reference / UWB anchors. Besides, a CDF will be also helpful to fully understand the error distribution.
Round 2
Reviewer 3 Report
Comments and Suggestions for Authors
The authors addressed my comments well.
Reviewer 4 Report
Comments and Suggestions for Authors
The new version looks good to me. It addresses all the concern of mine.